# Influence of Soft Phase and Carbon Nanotube Content on the Properties of Hierarchical AZ61 Matrix Composite with Isolated Soft Phase

**DOI:** 10.3390/nano12162877

**Published:** 2022-08-21

**Authors:** Yunpeng Ding, Sijia Jiao, Yizhuang Zhang, Zhiai Shi, Jinbiao Hu, Xulei Wang, Zhiyuan Li, Hanying Wang, Xiaoqin Guo

**Affiliations:** School of Materials, Zhengzhou University of Aeronautics, Zhengzhou 450046, China

**Keywords:** carbon nanotube, hierarchical magnesium matrix composite, soft phase, microstructure, mechanical property

## Abstract

Carbon nanotube-reinforced magnesium matrix (CNTs/Mg) composite has great application potential in the transportation industry, but the trade-off between strength and ductility inhibits its widespread application. In order to balance the strength and plasticity of the composite, in this work, on the basis of the AZ61 matrix composite homogeneously reinforced by Ni-coated CNTs (hard phase), 30 vol.% large-size AZ61 particles are introduced as an isolated soft phase to fabricate hierarchical CNTs/AZ61 composites. The compression tests show the fracture strain and compressive strength of this composite increases by 54% and 8%, respectively, compared with homogeneous CNTs/AZ61 composite. During deformation, the hard phase is mainly responsible for bearing the load and bringing high strength, due to the precipitation of the Mg_17_Al_12_ phase, uniformly dispersed CNT and strong interfacial bonding of the CNTs/Mg interface through nickel plating and interfacial chemical reaction. Furthermore, the toughening of the soft phase results in high ductility. With the increase in CNT content, the compressive strength of composites is nearly unchanged but the fracture strain gradually decreases due to the stress concentration of CNT and its agglomeration.

## 1. Introduction

Magnesium alloy has great application potential in the field of aerospace, automobiles and electronics, due to its high specific strength and stiffness, good damping response, excellent machinability and good recyclability [1,2]. However, the limited absolute strength and ductility inhibits its widespread application [3]. The strength of magnesium matrix composites is greatly improved by adding reinforcement. Among reinforcements, carbon nanotubes (CNTs) are one of the most promising potential reinforcements [2] for magnesium matrix composites due to their advantages [1,4] such as high strength (~30 GPa), Young’s modulus (~1 TPa) and high aspect ratio. However, the plasticity of carbon nanotube-reinforced magnesium matrix (CNTs/Mg) composites declines sharply [5], and this trade-off between strength and plasticity greatly hinders their applications [6] and becomes a hot research topic.

To overcome this trade-off, a hierarchical (or heterogeneous) structure design has attracted more and more attention. Yeyang Xiang et al. [7] constructed CNTs/Mg micro-nano-layered composites by electrophoretically depositing a layer of CNT on Mg foils and subsequent rolling to imitate the structure of nacre. This composite shows a strengthening efficiency of 500 and a slight increase in toughness. Xi Luo et al. [6] embedded the elongated and curled pure Mg soft phase in the SiC/Mg hard phase to form an inverse nacre structure, resulting in high strength and high tensile elongation. Jinling Liu et al. [8] prepared hierarchical magnesium matrix composites reinforced by SiC nanoparticles by mechanical ball milling and spark plasma sintering, in which Mg and SiC particles are mixed to form a continuous hard phase, while pure Mg forms isolated the soft phase. Compared with pure magnesium, the compressive strength and ductility of this hierarchical composite are significantly improved. Furthermore, its ductility is also much higher than that of a homogeneous composite.

In the previous study by the authors [9], the particle size of the Mg matrix was significantly refined by long-time ball milling. Then, homogeneous CNTs/Mg composites were prepared by uniformly dispersing Mg particles with CNTs, which greatly improved the compressive strength and yield strength of this composite. However, its ductility had significantly decreased. In order to balance the relationship between the strength and plasticity of CNTs/Mg composites, in this paper, on the basis of the AZ61 matrix homogeneously reinforced by Ni-coated CNTs, large-size pure AZ61 particles are introduced as an isolated soft phase to prepare the hierarchical CNTs/AZ61 composite. The influence of the soft phase and CNT content in hard phases on the microstructure and mechanical properties of hierarchical CNTs/AZ61 composites was investigated.

## 2. Materials and Methods

The raw materials were AZ61 atomised powder (spherical, particle size 80 μm, purity 99.9 wt%, Nanou Co., Ltd., Shanghai, China) and Ni-coated multi-wall CNTs (length less than 5 μm, outer diameter of approximately 30–50 nm, inner diameter 5–12 nm, CNT content > 58 wt%, Ni content > 28 wt%, fabricated by electroless plating, Chinese Academy of Sciences Chengdu Organic Chemistry Co., Ltd., Chengdu, China). The preparation process of composite was mainly divided into three steps: ball milling grinding of AZ61 powder, ball milling mixing of AZ61 original powder, AZ61 refined powder and CNTs, and vacuum hot pressing sintering, as shown in Figure 1.

At the first step, as-received AZ61 powder was grinded by planetary ball milling process at a speed of 225 r/min for 30 h. Zirconia ball was used and the ratio of ball to material was 10:1. Stearic acid was employed as a control agent. The second step was the ball milling mixing to prepare the composite powder. The AZ61 original powder (volume fraction 30%), AZ61 refined powder and Ni-coated CNTs (with a volume fraction of 0.5%, 0.75%, 1.0% and 1.25%, respectively) into the sealable tank and then milled at a speed of 225 r/min for 4 h. The ball type and ball material ratio were the same as the previous step. The volume fraction of CNT was calculated from the mass fraction and density. In the third step, the composite powder was sintered by vacuum hot pressing at 500 °C for 1 h under 40 MPa. The heating rate was 10 °C/min.

The composite was cut into small samples for structural characterisation. The metallographic structure of composite was observed by metallographic microscope. The powder and composite materials were observed by scanning electron microscopy (SEM, JSM-7001F, JEOL, Tokyo, Japan). The interface structure of composites was observed by transmission electron microscope (TEM, TECNAI G2 F20S-TWIN, FEI company, Hillsboro, OR, USA). The composite was cut into 5 mm × 5 mm × 12.5 mm samples for compression test under a compression rate of 0.0625 mm/min. During the compression test, three specimens were used for each treatment level. The average value and standard deviation of the strength and fracture strain were calculated and analysed in detail. The stress and strain in compression were calculated using Equations (1)–(4).

Engineering stress
(1)$$\sigma_{E}=\frac{P}{A_{0}}$$

Engineering strain
(2)$$\varepsilon_{E}=\frac{\Delta l}{l_{0}}$$

True stress
(3)$$\sigma_{T}=\sigma_{E}\left(1-\varepsilon_{E}\right)$$

True strain
(4)$$\varepsilon_{T}=-ln\left(1-\varepsilon_{E}\right)$$

The meaning of the symbol is as follows: *P* applied load, *A*_0_ the original cross-sectional area, Δ*l* the change in length, and *l*_0_ the original length.

## 3. Results

### 3.1. Morphology of Powder

Figure 2a shows the morphology of the original spherical AZ61 powder with the particle size of 80 μm. After 30 h of ball milling, the AZ61 powder becomes irregular small particles with the size of 2–25 μm due to the violent collision, extrusion and friction during the ball milling process, as shown in Figure 2b. The refinement of AZ61 powder increases the surface area, which is conducive to the dispersion of CNT on the particle surface [9] and can improve the properties of the final composite through fine grain strengthening. Figure 2c,d are the microstructure of nickel-coated CNTs. CNTs are coated by fine nickel particles with a size lower than 50 nm. The distribution and content of the element in Ni-coated CNT from EDS are shown in Figure 3 and Table 1, respectively. It is mainly composed of C and Ni, which correspond to CNT and Ni particles, respectively. These nickel particles facilitate the interfacial bonding of CNTs with magnesium alloys [9].

Figure 4 shows the SEM morphology of composite powders. It can be seen from Figure 4a,b that the composite powder consists of two parts: the coarse particles with a size of approximately 80 μm (original AZ61 powder) and the fine particles with a size of less than 20 μm (AZ61 fine powder after grinding). Figure 4c–f show the SEM morphology of fine particles in the composite powder with different CNTs contents. It is evident that the fine particles consist of AZ61 fine powder and CNTs distributed on its surface. These CNTs were unaligned and bend. Moreover, when the CNTs content is low, CNTs can be uniformly dispersed to the surface of AZ61 powder by ball milling, as shown in Figure 4c,d, by high-speed rotation to continuously squeeze, collide and stir [10] the mixed powder. As the amount of CNT increases, the distribution of CNTs on the surface of AZ61 powder also increases. When the content of CNTs is higher than 1 vol.%, CNTs cannot be effectively dispersed, and large agglomerations of CNTs appear in composite powder, as shown in Figure 4f.

### 3.2. Microstructure of Composite

Figure 5 shows the microstructure of hierarchical CNTs/AZ61 composites with different CNT contents. Those grey areas are hard phases derived from CNTs and small AZ61 particles refined by ball milling, and the white areas are soft phases formed by the large pure AZ61 particle. The continuous hard phase zone enriched with CNTs isolates the soft phase zone from each other. There are also many relatively small soft phases wrapped in the hard phase region. This part of the small soft phase is formed by welding the AZ61 powder impacted by zirconia ball during ball milling. Black spots appear in Figure 5b after the increase in CNT number, which just matched the agglomerations of CNT in the mixed powder (Figure 4).

Figure 6a–d show the SEM morphology of hierarchical 0.5 vol.% CNTs/AZ61 composite. It can also be observed that the composite consists of a continuous hard phase and separated soft phase, which is similar to the result in the microstructure diagram (Figure 5). At high magnification, the hard phase contains a large number of small reticulated phases (Figure 6d).

In order to clarify the composition of the hard phase and soft phase, energy-dispersive spectrometer (EDS) analysis was carried out. Figure 7 shows the EDS maps of hierarchical 0.5 vol.% CNTs/AZ61 composites. Compared with the soft phases, the enrichment of C, Ni and Al elements appeared in the hard facies. The C and Ni corresponds to Ni coated CNTs. Considering that the matrix is the AZ61 alloy, it can be concluded that Al corresponds to Mg_17_Al_12_ phase. This is because the Mg_17_Al_12_ phase is easy to precipitate at the grain boundary [11] and segregation occurs. During the process of grinding, a large number of grain boundaries (grain boundaries after sintering) were formed between the fine AZ61 particles in the hard phase region, which is the ideal place for Mg_17_Al_12_ precipitation.

Figure 8 shows the TEM diagrams of a hierarchical 1 vol.% CNTs/AZ61 composite. It can be seen that the composite consists of a continuous hard phase and independently distributed soft phase (Figure 8a). A large number of uniformly distributed CNTs can be seen in the enlarged view of the hard phase (Figure 8b). These CNTs were unaligned and bent. There is an MgNi_2_ phase near the CNTs/Mg interface (Figure 8c,d) which is caused by the chemical reaction between the Ni layer of the CNT and the matrix [12]. It is these moderate chemical reactions that make the CNT have better interfacial bonding with a magnesium alloy matrix, which is conducive to the improvement of properties.

### 3.3. Mechanical Property

Figure 9 shows the compressive stress–strain curve and the change trend of mechanical properties of hierarchical CNTs/AZ61 composites and its matrix. The data show that adding CNTs into the AZ61 matrix with hierarchical structure can greatly improve its strength. The standard deviation of these data (Figure 9b) is very small, which means that the statistical variability is little. When the CNTs content is 0.5 vol.%, the compressive strength and yield strength of composites increase from 330 MPa and 220 MPa to 441 MPa and 306 MPa, respectively, and rise by 34% and 39%, respectively. However, the fracture strain slightly decreases by only 3% (from 12.1% to 11.7%). With the increase in the CNT content, the compressive strength of composites changes little but the fracture strain decreases gradually. Compared between the performance of CNTs/AZ61 composites in this study and the similar magnesium matrix composites reported in the literature [13,14,15], the composite has advantages in mechanical properties.

In order to characterise the performance differences between homogeneous and hierarchical composites during ball milling, the compressive stress–strain curves of AZ61, homogeneous and hierarchical 0.5 vol.% CNTs/AZ61 composites were presented, as shown in Figure 10. Compared with the homogeneous CNTs/AZ61 composite, the fracture strain of the hierarchical composite significantly increased by 54% (from 7.6% to 11.7%) and the compressive strength rose by 8% (from 409 MPa to 441 MPa).

### 3.4. Fracture Morphology

Figure 11 shows the fracture morphology in homogeneous and hierarchical CNTs/AZ61 composites. At the fracture of homogeneous composite (Figure 11a,b), many adherent particles can be seen, whose size are equivalent to that of the refined particle after ball milling in Figure 2b. Therefore, it is proven that the fracture spreads along the particle interface (grain boundary). However, the fracture characteristics of hierarchical composites were mixed. In the fracture surface of hierarchical composite (Figure 11c,d), in addition to adhesion particles, a number of relative flat tearing regions (inside the blue coil) can be seen. Its size is comparable to that of the soft phase. It is an indication of the quasi-cleavage fracture of the soft phase due to the hexagonal close-packed structure of Mg [13,14,15]. Obviously, the cracking modes are significantly different between homogeneous and hierarchical CNTs/AZ61 composites.

Figure 12 illustrates the distribution of CNTs in the fracture surfaces of hierarchical CNTs/AZ61 composites with different CNT contents. When the content of CNTs is small, CNTs can be well dispersed by ball milling, and single CNTs Are pulled out at the fracture with a work of pullout, which, to the order of magnitude, can be identified with ptr_0_L^2^ [16], where t is the CNT/matrix interfacial shear stress, r_0_ is the radius of the CNT and L is the length of the CNT. With 1 vol.% of CNTs, locked CNTs could be observed at the fracture (Figure 12c). When the CNT content reached 1.25 vol.%, large aggregates of CNTs could be found at the fracture (Figure 12d).

## 4. Discussion

In the first step of the preparation process for the hierarchical CNTs/AZ61 composite, the grinding process makes the as-received AZ61 particles significantly finer (Figure 2b). The role of this process is two-fold. On the one hand, it refines the grains in the hard phase of composite (Figure 5 and Figure 6), resulting in fine-grain strengthening according to the Hall–Petch relationship. On the other hand, it increases the surface area of the matrix particle and provides more space for the dispersion of CNT [9]. In the second step, the CNTs, refined AZ61 particles and as-received AZ61 particles were evenly dispersed together in the ball milling dispersion process (Figure 4). After sintering, the final composite consists of two parts (Figure 6, Figure 7 and Figure 8): the hard phase with high strength enriched in CNTs and Mg_17_Al_12_ phases and the soft phase without CNT with relatively good ductility. In composite, those dispersed CNTs were unaligned and bent (Figure 4 and Figure 8). Even if the CNT is bent, the path and interface of deformation and fracture in the composite can still be increased during deformation and fracture, so that the energy required for fracture can be greatly increased, and finally play an important role in strengthening the composite. As for composites reinforced by CNTs, the nominal tensile strength of CNT is an important design factor in determining the mechanical properties and that the nominal (engineering) tensile strength is a product of the fracture strength (effective strength) and fracture cross-section ratio that can be calculated by the fracture cross-sectional area divided by the full cross-sectional area including the hollow core [4]. The nominal tensile strength of the MWCNTs is approximately 10.8 ± 6.9 GPa [4], which is much higher than the matrix resulting in strengthening. In particular, the nickel coating on the surface of CNTs reacts with Mg to generate a certain amount of MgNi_2_ phases (Figure 8) at the interface during sintering, which enables CNT to obtain a good interface combination with the matrix [9]. From the fracture morphology (Figure 12), the soft phase surface is characterised by tearing fracture morphology, while the hard phase presents the fracture morphology extending along the boundary of grain or particle, which proves that the hard phase has poor plasticity and it generally indicates brittle fracture characteristics [9]. This result is similar to the previous result [9] of the authors. The strength and ductility of the hierarchical composite are obviously better than that of the homogeneous composite (Figure 10). During deformation, the hard phase is mainly responsible for bearing the load and bringing high strength [6], but the CNT and its aggregates distributed in the hard phase (Figure 4, Figure 5 and Figure 12) easily cause stress concentration and produce microcracks. The premature failure of CNT could occur and this could lead to stress concentration and microcrack initiation in the matrix. Premature failure could be caused by the presence of residue strain in the deformed CNT (Figure 3 and Figure 7). Additionally, defects in CNT, such as Stone–Wale defects, could affect the mechanical response of the CNT. In particular, the presence of these defects could facilitate the interaction with one another to influence the energy uptake of the deforming CNT during axial reinforcement (when the applied load acts axially on the CNT) [17] and torsional reinforcement (when the applied load acts to twist the bended CNT) [18]. These defects could lead to premature fracture of the CNT, triggering stress concentration and microcracks initiation. However, when the microcracks spread to the surrounding soft phase during the process of propagation, the high ductility of the soft phase makes the stress release and inhibit the propagation of cracks [6]. As a result, the soft phase becomes the final fracture zone. However, the plasticity of a soft phase is limited due to the hexagonal close-packed structure of magnesium alloy, so the fracture surface of the soft phase is relatively flat [19] and an indication of the quasi-cleavage fracture [20]. In short, the hard phase is responsible for sustaining loads and thus increasing strength, while the soft phases play an important role in inhibiting crack propagation and thus enhancing plasticity. Therefore, the strength and plasticity of a hierarchical composite are higher than those of the uniform composite (Figure 10).

In a CNT-reinforced metal matrix composite, the load transfer effect through CNTs is the dominant strengthening mechanism [21]. When the CNT content is high, the load transfer effect of reinforcement is enhanced and the strength increases. On the other hand, the number of points for stress concentration also increases, which is not good for strength. Those two effects compete with each other and result in nearly unchanged strength (Figure 9). When the CNT content is high, the stress concentration can more easily occur, especially after agglomeration, which leads to cracking. Therefore, with the increase in CNT content, the ductility of the composites becomes worse (Figure 9).

## 5. Conclusions

In order to balance the relationship between the strength and plasticity of homogeneous CNTs/Mg composites, in this work, on the basis of refined AZ61 particles, large-size pure AZ61 particles are introduced as the soft phase in hierarchical CNTs/AZ61 composites. Compared with the homogeneous CNTs/AZ61 composite, the fracture strain and compressive strength of this composite significantly increases by 54% and 8%, respectively.

During deformation, the hard phase is mainly responsible for bearing the load and bringing high strength, which is due to the precipitation of the Mg_17_Al_12_ phase, uniformly dispersed CNT and strong interfacial bonding of CNTs/Mg interface through nickel plating and interfacial chemical reaction. On the other hand, the soft phase inhibits the propagation of the crack, resulting in high ductility.

With the increase in CNTs content, the compressive strength of composites changes little but the fracture strain gradually decreases due to the stress concentration of CNT and its agglomeration.

## Figures and Tables

**Figure 1 nanomaterials-12-02877-f001:**
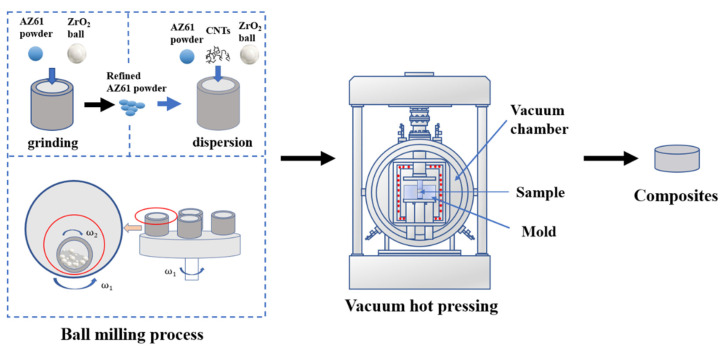
Preparation flow chart of hierarchical/heterogeneous CNTs/AZ61 composites.

**Figure 2 nanomaterials-12-02877-f002:**
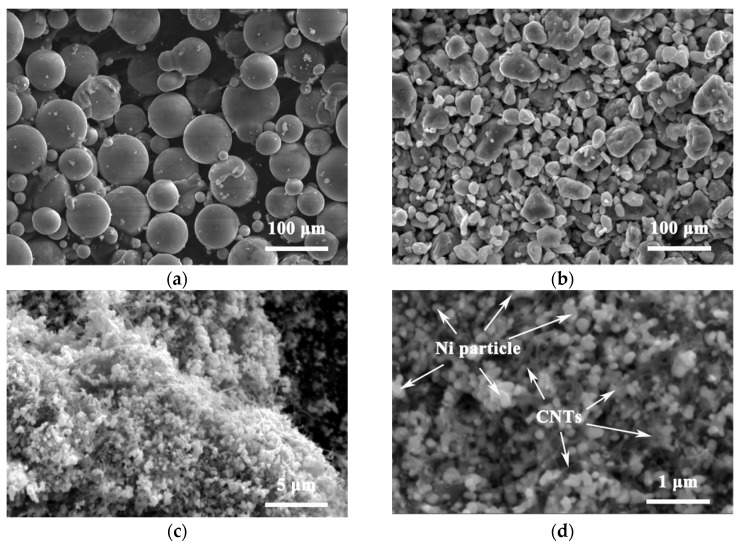
SEM diagrams of AZ61 powder: (**a**) original and (**b**) after grinding of ball milling. SEM images of Ni-coated CNTs with (**c**) low magnification and (**d**) high magnification.

**Figure 3 nanomaterials-12-02877-f003:**
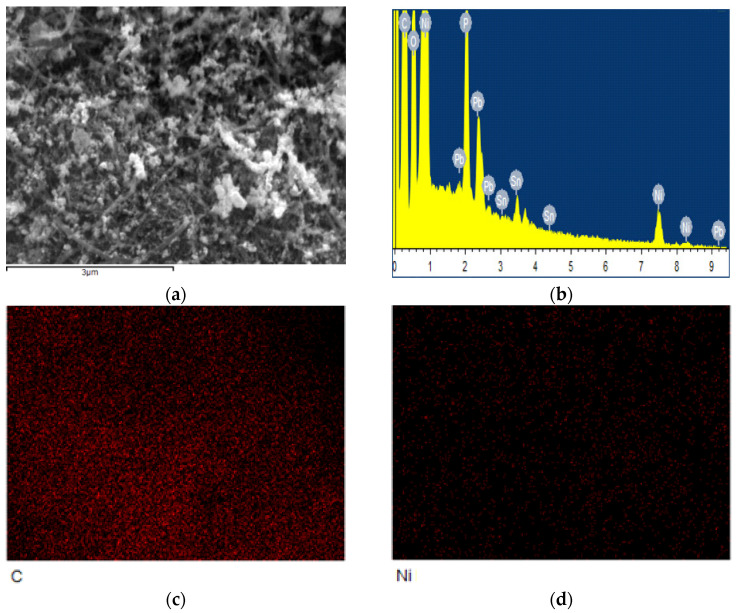
The distribution of element in Ni-coated CNT from EDS: (**a**) SEM diagram; (**b**) EDS pattern; (**c**) C element; and (**d**) Ni element.

**Figure 4 nanomaterials-12-02877-f004:**
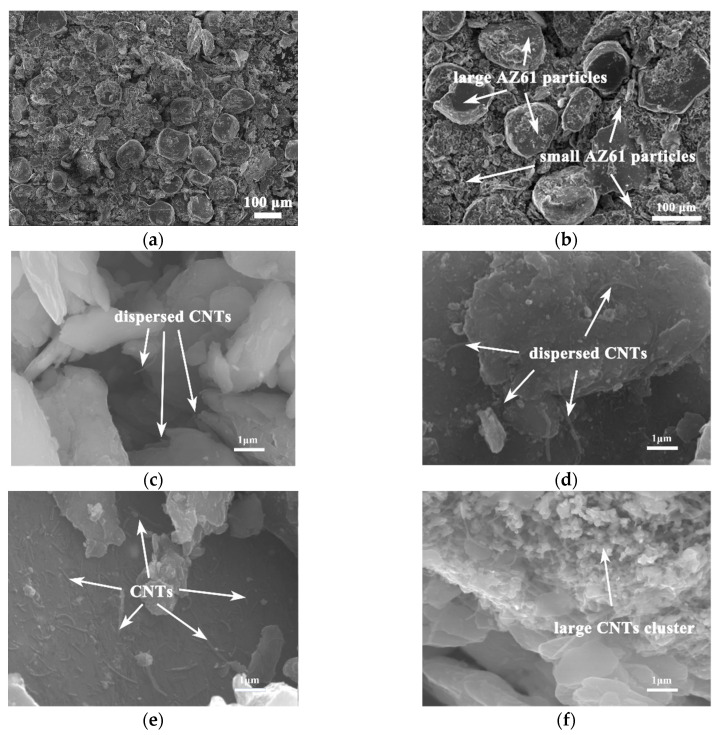
Typical low-magnification (**a**,**b**) and high-magnification SEM morphology of CNTs/AZ61 composite powder dispersed by ball milling under different contents of CNT. (**c**) 0.5%; (**d**) 0.75%; (**e**) 1%; and (**f**) 1.25%.

**Figure 5 nanomaterials-12-02877-f005:**
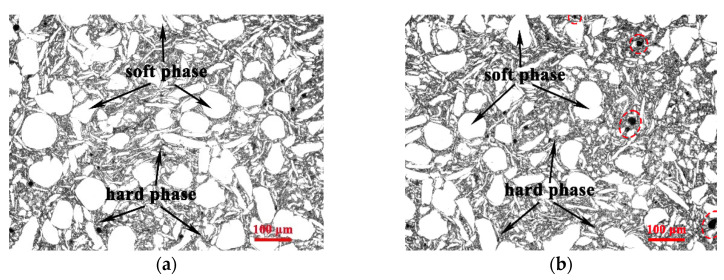
Microstructure of hierarchical CNTs/AZ61 composites with different magnifications: (**a**) 0.5 vol.%; and (**b**) 1.25 vol.%.

**Figure 6 nanomaterials-12-02877-f006:**
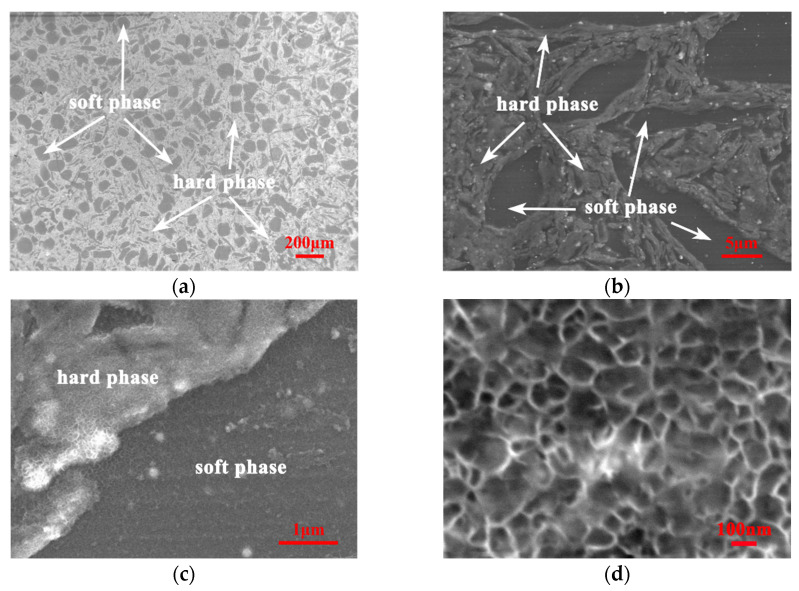
SEM images of hierarchical 0.5 vol.% CNTs/AZ61 composites with different magnifications, (**a**) ×50, (**b**) ×2000, (**c**) ×10,000 and (**d**) ×30,000.

**Figure 7 nanomaterials-12-02877-f007:**
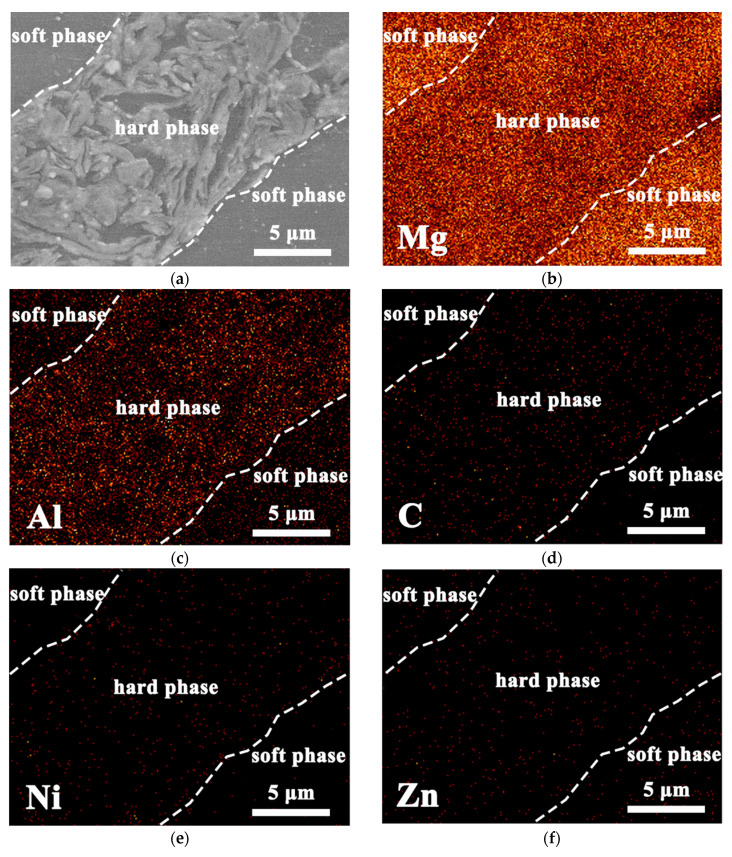
EDS maps of hierarchical 0.5 vol.% CNTs/AZ61 composites: (**a**) morphology; (**b**) Mg; (**c**) Al; (**d**) C; (**e**) Ni; and (**f**) Zn.

**Figure 8 nanomaterials-12-02877-f008:**
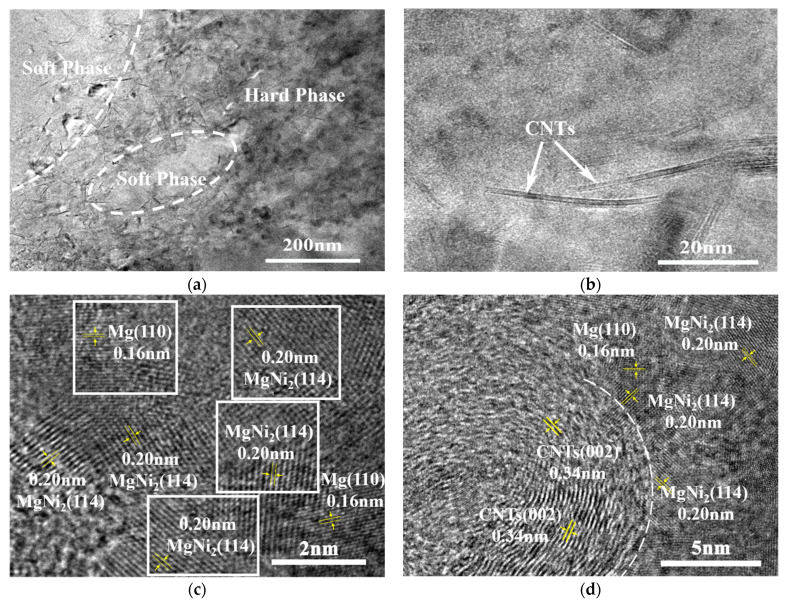
HRTEM diagrams of hierarchical 1 vol.% CNTs/AZ61 composites (**a**) hard phase and soft phase, (**b**) CNT distribution in hard phase, (**c**,**d**) CNT interface.

**Figure 9 nanomaterials-12-02877-f009:**
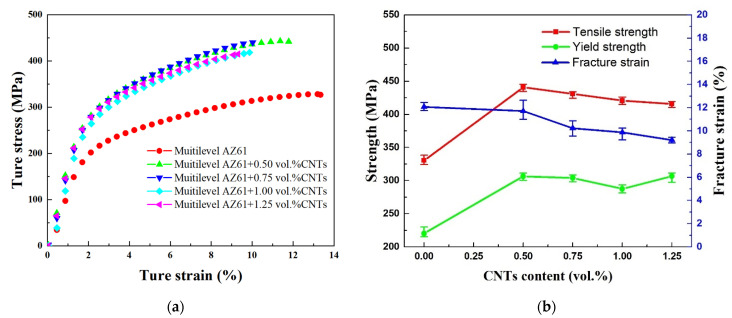
Typical compressive stress–strain curves (**a**) and variation curves of compressive strength, yield strength and fracture strain (**b**) of hierarchical composites with different CNT contents.

**Figure 10 nanomaterials-12-02877-f010:**
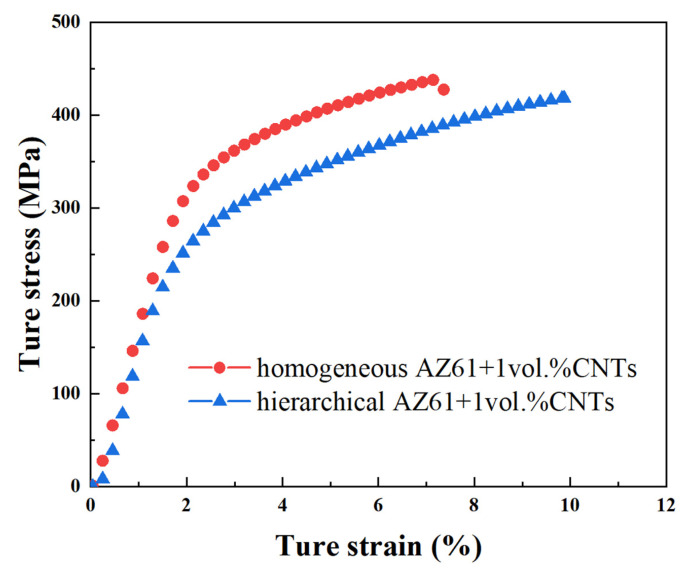
Typical compressive stress–strain curves of AZ61, homogeneous and hierarchical 0.5 vol.% CNTs/AZ61 composites.

**Figure 11 nanomaterials-12-02877-f011:**
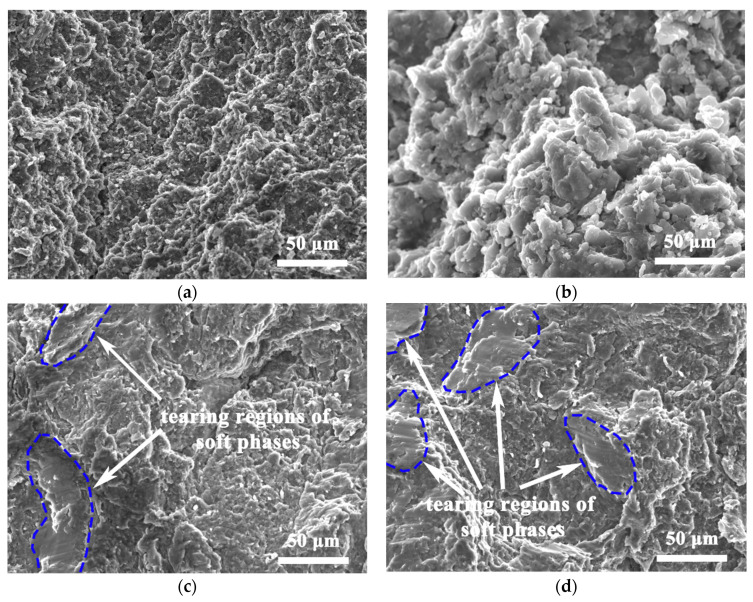
SEM images of fracture morphology in homogeneous (**a**,**b**) and hierarchical (**c**,**d**) CNTs/AZ61 composites.

**Figure 12 nanomaterials-12-02877-f012:**
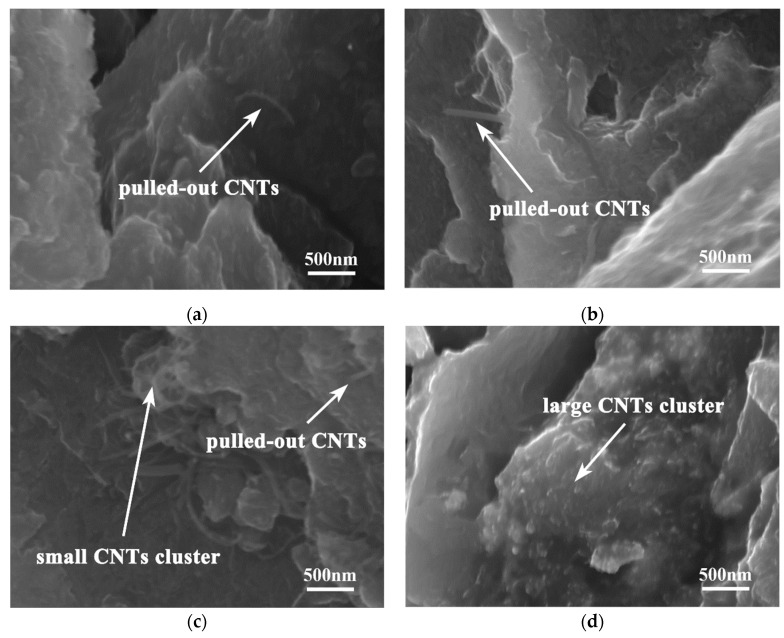
Distribution of CNTs in the fracture surfaces of hierarchical CNTs/AZ61 composite with different CNT contents: (**a**) 0.5 vol.%; (**b**) 1 vol.%; (**c**) 1.5 vol.%; and (**d**) 2 vol.%.

**Table 1 nanomaterials-12-02877-t001:** Element content from EDS of Ni-coated CNT.

**Element**	**Mass Fraction/%**	**Atomic Fraction/%**
C	57.67	82.43
Ni	28.47	8.32
--	13.86	9.25

## Data Availability

The resulting data are available to download from (https://pan.baidu.com/s/1WJnHW0Pqgw_4JiHGQGnQKw, accessed on 18 August 2022) with password of “l3t8”.

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
