# Peer review of "Influence of Soft Phase and Carbon Nanotube Content on the Properties of Hierarchical AZ61 Matrix Composite with Isolated Soft Phase"

_nanomaterials, 2022, doi:10.3390/nano12162877_

Round 1

Reviewer 1 Report

This paper is about hierarchical matrix composite materials. As I understood, in this paper, large-size pure AZ61 particles are introduced as isolated soft phase to prepare hierarchical CNTs/AZ61 composite, where the CNT provides the reinforcement phase. The microstructure and mechanical properties of hierarchical CNTs/AZ61 composites were investigated for varying influence of Ni-coated CNT content.

Here are my comments and suggestions.

Line 58-59: Specify what type of CNT was used: Single or multi-wall CNT?

Figure 3 (and Figure 7): I would like to acknowledge the authors' attempt to produce the excellent/clear photos of CNTs embedded in composite. It is interesting to see CNTs dispersed, and bend; it suggests that this could lead to residue stress and energy absorbed to cause the deflection. Authors could explain how these unaligned, bend CNTs could help provide reinforcement to the composite materials.

As observed in Figure 8, since it is not known how many specimens were used for each treatment level, it is difficult to see how the strength and fracture strain (mean +- standard deviation of mean) vary statistically (if any) for each CNT’s content %. Perhaps the authors could comment on the (absence of) statistical variability issue in the strength and fracture strain, particular with regard to Figure 8 b.

Line 214-215: " When the content of CNTs is small, 214 CNTs can be well dispersed by ball milling, and single CNTs is pulled out at the fracture. ". For the benefit of capturing a wider readership relevant to this work, here is a suggestion for the authors to change to: "When the content of CNTs is small, 214 CNTs can be well dispersed by ball milling, and single CNTs is pulled out at the fracture with a work of pullout, which to order of magnitude, can be identified with ptr0L2 [Ref 0], where t is the CNT/matrix interfacial shear stress, r0the radius of the CNT and L the length of the CNT."

where,

[Ref 0] "Influence of fibre taper on the work of fibre pull-out in short fibre composite fracture", Journal of Materials Science (2010) 45:1086–1090, DOI 10.1007/s10853-009-4050-2

Line 248-250: "During deformation, the hard phase is mainly responsible for bearing the load and 248 bringing high strength [4], but the CNT and its aggregates distributed in hard phase 249 (Figure 3, 4 and 11) easily cause stress concentration and produce microcracks". For the benefit of capturing a wider readership relevant to this work, I would like to suggest to the authors to append the following statements after this sentence as follows: "During deformation, the hard phase is mainly responsible for bearing the load and 248 bringing high strength [4], but the CNT and its aggregates distributed in hard phase 249 (Figure 3, 4 and 11) easily cause stress concentration and produce microcracks. Premature failure of CNT could occur and this could lead to stress concentration and microcrack initiation in the matrix. Premature failure could be caused by the presence of residue strain in the deformed CNT (Fig. 3 and Fig. 7).  Additionally, defects in CNT, such as Stone-Wale defects, could affect the mechanical response of the CNT. In particular, the presence of these defects could facilitate interaction with one another to influence the energy uptake of the deforming CNT during axial reinforcement (when the applied load acts axially on the CNT) [Ref 1] and torsional reinforcement (when the applied load acts to twist the bended CNT) [Ref 2]. These defects could lead to premature fracture of the CNT, triggering stress concentration and microcracks initiation."

where,

[Ref1]: “On defect interactions in axially loaded single-walled carbon nanotubes”, Journal of Applied Physics 103, 054306 (2008); https://doi.org/10.1063/1.2837835

[Ref 2]: “Defect-defect interaction in single-walled carbon nanotubes under torsional loading”, International Journal of Modern Physics B, Vol. 24, No. 10, pp. 1215-1226 (2010); https://doi.org/10.1142/S021797921005510X

Reviewer 2 Report

See attachment

Round 2

Reviewer 2 Report

Necessary amendments have been made in the revised manuscript.